

# A retrospective study: cardiac MRI of fulminant myocarditis in children—can we evaluate the short-term outcomes?

Haipeng Wang[1], Bin Zhao[1], Haipeng Jia[2], Fei Gao[1], Junyu Zhao[3] and Cuiyan Wang[1]

[1] Shandong Medical Imaging Research Institute Affiliated to Shandong University, Ji'nan, China
[2] Department of Radiology, Qilu Hospital of Shandong University, Ji'nan, China
[3] Department of Internal Medicine, Shandong Provincial Qianfoshan Hospital, Ji'nan, China

Corresponding author
Cuiyan Wang, cywang0729@163.com

## ABSTRACT

**Background.** Fulminant myocarditis (FM) is an inflammatory disease of the myocardium that results in ventricular systolic dysfunction and causes acute-onset heart failure. Cardiac magnetic resonance (CMR) has become the primary noninvasive tool for the diagnosis and evaluation of myocarditis. The aim of our study was to assess the CMR findings at different course of FM and the short-term outcomes of fulminant myocarditis (FM) in children.

**Methods.** Eight FM children with CMR examinations were included in our study. Initial baseline CMR was performed 10 days (range, 7–20 days) after onset of FM and follow-up CMR after 55 days (range, 33–75 days). Cardiac morphology and function and myocardial tissue characterization at baseline and follow-up CMR were compared using paired $T$-test and Mann–Whitney $U$ test. The clinical data and initial CMR findings were also compared to predict short-term outcomes.

**Results.** The median age of eight FM children was 8.5 years old (range, 3–14). The initial CMR findings were most common with early gadolinium enhancement (EGE, 100%), followed by signal increasing on T2WI and late gadolinium enhancement (LGE, 87.5%), increased septal thickness (75.0%) and increased left ventricle ejection fraction (LVEF, 50.0%). Only three LGE (37.5%), one signal increasing on T2WI (12.5%) and one increased LVEF (12.5%) were found at follow-up. Statistically significant differences were found between initial and follow-up CMR abnormalities in the septal thickness, left ventricular end-diastolic diameter (LVEDD), end-systolic volume (ESV), LVEF, left ventricular mass, T2 ratio and LGE area ($P = 0.011, P = 0.042, P = 0.016, P = 0.001, P = 0.003, P = 0.011, P = 0.020$). The children with full recovery performed higher incidence of III° atrioventricular block (AVB, five cases VS 0 case) and smaller LGE area ($104.0 \pm 14.5$ mm$^2$ VS $138.0 \pm 25.2$ mm$^2$) at baseline CMR.

**Discussion.** The CMR findings of FM in children were characteristic and useful for early diagnosis. Full recovery of clinical manifestations, immunological features and CMR findings could be found in most FM children. The presence of III° AVB and smaller LGE area at baseline CMR might indicate better short-term outcomes.

## INTRODUCTION

Fulminant myocarditis (FM) is an inflammatory disease of the myocardium that results in ventricular systolic dysfunction and causes acute-onset heart failure (*Gupta et al., 2008*; *Felker et al., 2000*; *Ginsberg & Parrillo, 2013*). Despite the initial severe presentation, excellent short-term and long-term outcomes of FM with complete recovery of clinical manifestations and cardiac function have been found in the adult population (*Felker et al., 2000*; *Ginsberg & Parrillo, 2013*; *McCarthy et al., 2000*). While in the pediatric population, the outcomes of FM are controversial, with the mortality varied from 9.1% to 48.4% (*Amabile et al., 2006*; *Saji et al., 2012*; *Lee et al., 2014*).

The Dallas criteria, the standardized histologic criteria, have been identified as the gold standard in the diagnosis of myocarditis (*Cooper et al., 2007*). Because of the disadvantages of invasive procedures, sampling errors, severe complications and poor inter-observer agreement (*Chow et al., 1989*; *Hauck, Kearney & Edwards, 1989*; *Shirani, Freant & Roberts, 1993*), its application was only limited when it might alter management or offer a meaningful prognosis (*Cooper et al., 2007*).

Nowadays, cardiac magnetic resonance (CMR) has become the primary noninvasive tool for the diagnosis and evaluation of myocarditis (*Bruder et al., 2009*). The diagnostic CMR criteria for myocarditis, Lake Louise Criteria, have been proposed with a diagnostic accuracy of 78% (*Friedrich et al., 2009*). However, to our knowledge, few studies have reported the CMR findings of FM in adults, and scarce in children. Besides, the recovery of myocardial tissue characterization was seldom considered in the outcomes of FM.

The aim of our study was to assess the CMR findings at different course of FM in children, to evaluate the short-term outcomes of FM from the aspect of clinical manifestations immunological features and initial CMR findings, and to find out the predictors of the short-term outcomes of FM.

## MATERIALS & METHODS

In this retrospective study, all FM children with CMR examinations in Shandong Provincial Hospital from January 2010 to December 2015 were assessed and eight children with initial and follow-up CMR examinations were enrolled in our study. Written informed consent was obtained from the parents of FM children. The median age of eight children was 8.5 years old (range, 3–14). All eight FM children were clinically diagnosed by an experienced pediatrician who had more than 30 years working experience and satisfied following criteria: (1) Recent history of viral prodromata with fever lasting < 2 weeks; (2) Acute onset (time interval between onset and hospital admission < 7 days); (3) Severe heart failure (at minimum requiring intravenous inotropic support), cardiac shock or severe arrhythmia; (4) Left ventricular dysfunction assessed by echocardiography; (5) Exclusion criteria: acute or chronic myocarditis, other non-ischemic cardiomyopathy, congenital heart disease, myocardial infarctions, cardiac tumors, autoimmune disease or extra-cardiac diseases that could explain the clinical manifestations (*Lieberman et al., 1991*; *Ramachandra et al., 2010*). Clinical manifestations immunological features, viral serology electrocardiography and initial and follow-up CMR findings and treatment of eight children were recorded. The

study protocol was approved by the institutional ethics committee of Shandong Medical Imaging Research Institute (NO. 2016-001).

## CMR imaging protocol

CMR imaging was performed with a clinical 3.0-T MR scanner (Achieva 3.0T TX; Philips Healthcare, Best, The Netherlands) equipped with dual-source parallel radiofrequency transmission. A 16-channel torso phased-array receiver coil was used for signal reception. All data acquisition was retrospective ECG gated. Children (>7 years) who could control breathing would acquire data with respiratory gating. Small children (<7 years) would be sedated with 10% chloral hydrate and examined under free-breathing condition. Multiple signal averaging were applied in children with free-breathing scanning to ''average out'' the respiratory motion and the through plane motion artifacts (*Krishnamurthy et al., 2015*; *Abd-Elmoniem et al., 2011*).

The CMR imaging protocols included cine imaging, T2-weighted imaging, early gadolinium enhancement (EGE) and late gadolinium enhancement (LGE). Cine images in three long axis (4-chamber, 2-chamber and 3-chamber) and sequential short axis (SA) from ventricular base to apex were acquired with balanced steady state free precession (b-SSFP) sequences. Imaging parameters were: repetition time (TR), 3.0 ms; echo time (TE), 1.5 ms; flip angle (FA), 45°; field of view (FOV), $220 \times 280$ mm$^2$; matrix, $200 \times 256$; slice thickness: 8 mm without slice gap; number of signal averages (NSAs), 1 (breath-hold children) or 3 (free-breathing children). T2-weighted images were acquired in 4-chamber and SA with a triple inversion recovery fast spin echo sequence to evaluate myocardial edema. Imaging parameters were: TR, $2 \times$ beats ms; TE, 60 ms; FA, 90°; FOV, $220 \times 280$ mm$^2$; matrix, $208 \times 167$; slice thickness: 8 mm. Pre- and post-T1 fast spin echo (FSE) imaging were acquired in 4-chamber before and after an intravenous bolus of 0.1 mmol/kg Gd-DTPA (Magnevist, Bayer, Germany) within 3 min. T1 FSE imaging parameters were: TR, $1 \times$ beats ms; TE, 10 ms; FA, 90°; FOV, $220 \times 280$ mm$^2$; matrix, $228 \times 171$; slice thickness: 8 mm. After EGE, another 0.1 mmol/kg Gd-DTPA was injected and LGE were performed in three long axis and sequential SA7-10 min later, using an a 3D phase-sensitive inversion recovery (3D-PSIR) sequence. Imaging parameters were: TR, 5.3 ms; TE, 2.5 ms; FA, 25°; FOV, $220 \times 280$ mm$^2$; matrix, $156 \times 150$; slice thickness: 8 mm; NSAs, 1 (breath-hold children) or 3 (free-breathing children).

## CMR imaging analysis

The left ventricle (LV) morphology and cardiac function were quantitatively evaluated on the cine images with the work station (EWS; Philips Healthcare). Two experienced CMR radiologists assessed the CMR findings independently and consensus was reached. The endocardial and epicardial contours of LV in the sequential SA of cine images were adjusted manually and the following LV cardiac functional parameters were automatically acquired: the end-diastolic volume (EDV), end-systolic volume (ESV), ejection fraction (EF), and LV mass. The papillary muscles and trabeculations were included as part of LV mass (*Buechel et al., 2009*). Impaired LV systolic function were considered if EF < 50% and enhanced LV systolic function if EF > 70%. Left ventricular end-diastolic diameter (LVEDD) was

measured in 4-chamber. The end-diastolic mid-septal thickness was measured and averaged in 4-chamber and SA of cine images. Increased septal thickness was considered by visual comparison at different stages of FM.

In 4-chamber and SA of T2 weighted images regional or global signal intensity of LV myocardium was measured as well as the signal intensity of subscapularis muscle in the same slice, excluding high signal of inadequately suppressed slowly flowing cavitary blood. The global signal intensity of LV myocardium as well as subscapularis muscle in 4-chamber of pre- and post-T1 weighted images were also measured. Then, the T2 ratio and EGE ratio were calculated according to Lake Louise Criteria (*Friedrich et al., 2009*). Myocardial edema and hyperemia were considered if T2 ratio > 1.9 and EGE ratio > 4.0. The segments, patterns and areas of myocardial necrosis/fibrosis on the LGE images were assessed and recorded using the 17-segment American Heart Association (AHA) model (*Cerqueira et al., 2002*).

### Short-term outcomes

The short-term outcomes of FM in children were evaluated according to clinical manifestations, immunological features and CMR findings at follow-up CMR. Poor short-term outcomes were defined as persistent clinical manifestations, abnormal levels of myocardial enzymes and CMR abnormalities within three months. Then the clinical data and imaging findings at baseline CMR of FM were evaluated to find out the predictors of short-term outcomes of FM.

### Statistical analysis

Categorical data were reported as percentage, and continuous data as the mean $\pm$ standard deviation (SD) or median (range). The normality of the variables was assessed by the Shapiro–Wilk test. Paired $T$-test and Mann–Whitney $U$ test were used to compare the intervals from onset to CMR examinations, the levels of cardiac troponin T (cTnT) and brain natriuretic peptide (BNP), cardiac morphology and function parameters and myocardial tissue characterization (T2 ratio, EGE ratio and LGE) at baseline and follow-up CMR of FM. All statistical tests were two-sided, and $P$-values less than 0.05 were considered as statistical significant. The clinical data and initial CMR findings between FM children with different short-term outcomes were directly compared because of small sample size.

The statistical analysis was carried out using SPSS version 19.0 (SPSS INC., Chicago, Illinois).

## RESULTS

### Patient characteristics

Patient characteristics and clinical data of FM at presentation were showed in Table 1. Viral serology was showed in Table S1. Echocardiography was performed within 72 h of the onset and the LVEF of FM children were 36.5 $\pm$ 8.2%. The clinical manifestations of FM were varied, including heart failure (eight cases, 100.0%), cardiac shock (four cases, 50.0%) and Adams-Stokes syndrome (four cases, 50.0%). ST-T changes in the electrocardiographic (ECG) examinations were observed in all FM children (100%), followed by III° atrioventricular block (AVB, 62.5%). All children showed abnormal

**Table 1** Characteristics and clinical data of eight FM children at presentation.

| Characteristics | N (%)/median(range) |
|---|---|
| Sex (M) | 5 (62.5%) |
| Median age (years) | 8.5 (3, 14) |
| LVEF (%) | 36.5 ± 8.2 |
| **Clinical manifestations** | |
| General symptoms | 5 (62.5%) |
| Fatigue | 5 (62.5%) |
| Fever | 1 (12.5%) |
| Gastrointestinal symptoms | 8 (100%) |
| Nausea and vomiting | 6 (75.0%) |
| Abdominal pain | 5 (62.5%) |
| Cardiovascular sysptoms | 8 (100%) |
| Heart failure | 8 (100%) |
| Cardiac shock | 4 (50.0%) |
| Chest pain/distress | 3 (37.5%) |
| Neurological symptoms | 5 (62.5%) |
| Headache/dizziness | 4 (50.0%) |
| Seizure | 4 (50.0%) |
| **Abnormal electrocardiogram** | |
| ST-T changes | 8 (100%) |
| III° AVB | 5 (62.5%) |
| **The highest levels of myocardial enzymes** | |
| cTnT (pg/ml) | 1711.5 (129.4, 6457.0) |
| BNP (pg/ml) | 11426.5 (546.1, 31648.0) |
| **Treatment** | |
| Intravenous immunoglobulin | 8 (100%) |
| Steroids | 8 (100%) |
| Isoprenaline | 5 (62.5%) |
| Dopamine | 3 (37.5%) |
| Ventricle-assist device | 2 (25.0%) |

**Notes.**

Values are presented as N (%) or median (range).

AVB, atrioventricular block; LVEF, left ventricle ejection fraction; cTnT, cardiac troponin T, reference limit: 3.0–14.0 pg/ml; BNP, brain natriuretic peptide, reference limit: 0–125.0 pg/ml.

levels of cTnT (1711.5 pg/ml, 129.4–6457.0 pg/ml; reference limit: 3.0–14.0 pg/ml) and BNP (11426.5 pg/ml, 546.1–31648.0 pg/ml; reference limit: 0–125.0 pg/ml) in the course of FM. All eight children received intravenous immunoglobulin and methylprednisolone after admission. Inotropic support was required in children for severe haemodynamic compromise: isoprenaline in five children and dobutamine in three children. Ventricle-assist devices also were implanted in two children.

The median intervals from onset to initial and follow-up CMR examinations were 10 days (range, 7–20) and 55 days (range, 33–75). At initial CMR, the clinical manifestations of FM had been markedly improved. Abnormal levels of cTnT (79.7 ± 71.1 pg/ml) were observed in six children (75%) and BNP (443.6 pg/ml, 75.5–4498.0) in seven children (87.5%). ST-T

**Table 2** Comparison between clinical data and CMR findings at baseline and follow-up CMR.

| | Baseline($N = 8$) | Follow-up($N = 8$) | $P$ values |
|---|---|---|---|
| Heart rate (/min) | $78.9 \pm 14.0$ | $86.9 \pm 13.3$ | 0.162 |
| Interval (days) | 10(7, 20) | 55(33, 75) | <0.001[*] |
| cTNT (pg/ml) | $79.7 \pm 71.1$ | $12.5 \pm 5.5$ | 0.061 |
| BNP (pg/ml) | 443.6(75.5, 4498.0) | 55.2(1.0, 340.2) | 0.011[*] |
| Cardiac morphology and function | | | |
| IVST (mm) | $10.9 \pm 3.1$ | $8.1 \pm 1.2$ | 0.011[*] |
| EDV (ml) | $44.6 \pm 12.6$ | $50.6 \pm 16.5$ | 0.161 |
| ESV (ml) | $12.9 \pm 5.0$ | $18.2 \pm 7.4$ | 0.016[*] |
| LVEF (%) | $71.8 \pm 6.9$ | $64.9 \pm 7.1$ | 0.001[*] |
| LVM (g) | $48.9 \pm 15.6$ | $36.9 \pm 12.6$ | 0.003[*] |
| LVEDD (mm) | $36.8 \pm 4.4$ | $38.4 \pm 4.5$ | 0.042[*] |
| Myocardial tissue characterization | | | |
| T2 ratio | $2.03 \pm 0.15$ | $1.80 \pm 0.11$ | 0.011[*] |
| EGEr | $8.9 \pm 3.5$ | 3.5 | – |
| LGE segments | 21(15.4%) | 11(8.1%) | 0.060 |
| LGE areas (mm$^2$) | 113.5(0, 165.0) | 0(0, 125.0) | 0.020[*] |

**Notes.**
Values are presented as a mean $\pm$ SD, median (range) or $n$ (%).
[*]$P$ values < 0.05.
cTnT, cardiac troponin T; BNP, brain natriuretic peptide; IVST, inter-ventricular septal thickness; EDV, end-diastolic volume; ESV, end-systolic volume; LVEF, left ventricle ejection fraction; LVM, LV mass; LVEDD, left ventricular end-diastolic diameter; EGE, early gadolinium enhancement; LGE, late gadolinium enhancement.

abnormalities were also observed in seven FM children (87.5%). At follow-up CMR, the symptoms of all FM children recovered completely. Only one children showed abnormal level of BNP (340.0 pg/ml).

## CMR findings

The CMR findings of FM at baseline and follow-up CMR were shown in Table 2. Two children (ages, 3 and 6 years old) were sedated and examined under free-breathing condition and the others with respiratory gating.

At baseline CMR, the characteristic CMR findings were the increased myocardial thickness in six children ($11.8 \pm 2.9$ mm). Four children (50.0%) performed increased LVEF ($77.2 \pm 4.5$%) and others with normal LVEF. No obviously abnormal myocardial motion was found. Pericardial effusion was found in three children (37.5%). At follow-up CMR, myocardial thickness returned to normal and 1 children (12.5%) showed increased LVEF (79.0%). Pericardial effusion in three children disappeared. Compared with follow-up CMR, FM children at baseline CMR showed increased septal thickness ($P = 0.011$), increased LV mass ($P = 0.003$), higher EF ($P = 0.001$), smaller LVEDD ($P = 0.042$) and lower ESV ($P = 0.016$) (Fig. 1).

The myocardial tissue characterization of FM at baseline CMR were most common with increased EGE ratio ($8.9 \pm 3.5$) in five FM children (100%) Regional or global increased T2WI signal was found in seven FM children (87.5%), with mean T2 ratio of $2.03 \pm 0.15$. In the LGE images, regional mid-wall enhancement were found in six cases (75.0%), followed

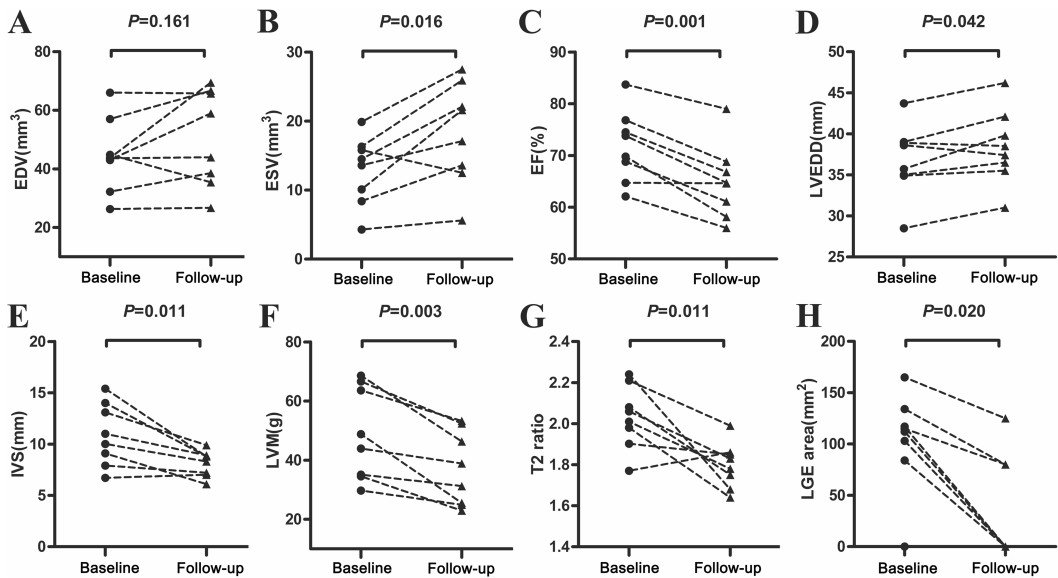

**Figure 1  Comparison of CMR findings in patients with FM at baseline and follow-up CMR.** (A) EDV, end-diastolic volume; (B) ESV, end-systolic volume; (C) EF, ejection fraction; (D) LVEDD, left ventricular end-diastolic diameter; (E) IVS, inter-ventricular septal thickness; (F) LVM, LV mass; (G) T2 ratio; (H) LGE area. Compared with the CMR findings of FM at follow-up, FM children at baseline CMR performed increased septal thickness ($P = 0.011$), LV mass ($P = 0.003$), EF ($P = 0.001$), T2 ratio ($P = 0.011$), LGE area ($P = 0.020$) and smaller LVEDD ($P = 0.042$) and ESV ($P = 0.016$).

by one subepicardial enhancement (12.5%). No predilections of myocardial segments were found for LGE. Diagnostic sensitivity of CMR abnormalities at baseline CMR for FM was 100% according to Lake Louise Criteria. At follow-up CMR only three children with decreased area of LGE (37.5%) and one child with increased T2WI signal (12.5%) were found. Statistically significant differences were found between initial and follow-up CMR in T2 ratio ($P = 0.011$) and LGE area ($P = 0.020$) (Fig. 1).

## Short-term outcomes

In our study, all children with FM survived. The short-term outcomes of FM in children were excellent. Full recovery of clinical manifestations, immunological features and CMR findings could be found in five FM children (62.5%) at follow-up CMR (Fig. 2). Abnormal levels of BNP and myocardial tissue characterization were also found in three children (37.5%). There were no significant differences of age, heart rate or intervals from onset to CMR examinations between FM children with and without full recovery.

Compared with three children without full recovery, the other five children showed higher incidence of III° AVB (5 cases VS 0 case), smaller LGE area ($104.0 \pm 14.5$ mm$^2$ VS $138.0 \pm 25.2$ mm$^2$) at baseline CMR of FM, which might indicate better short-term outcomes.

## DISCUSSION

In this study, we assessed the CMR findings at different course of FM in children. The CMR findings of FM in children were characteristic and useful for early diagnosis. Full recovery

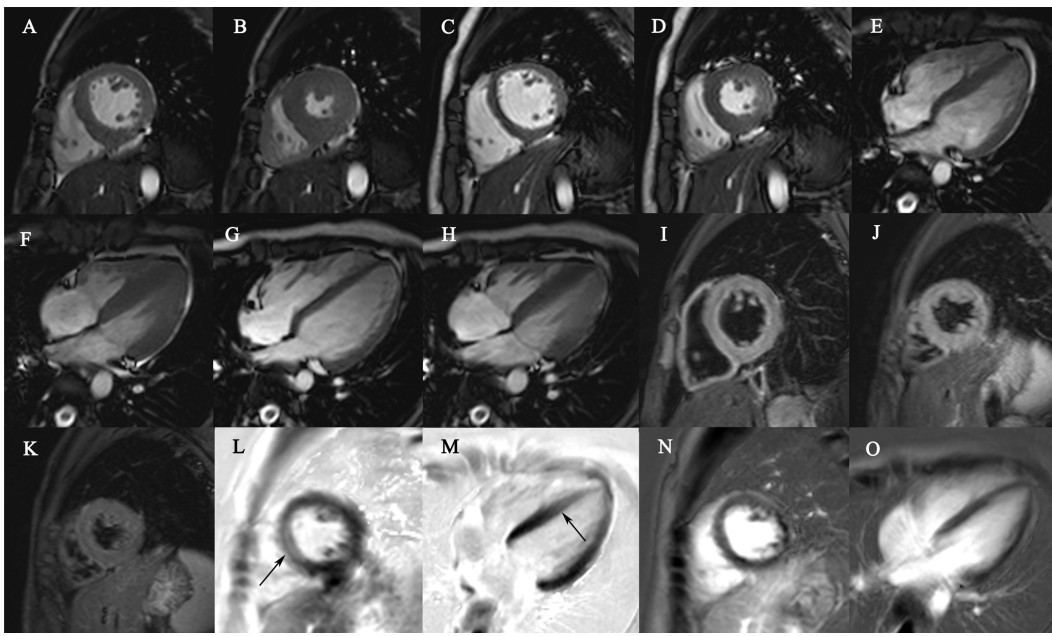

**Figure 2** Showed the CMR findings of a 13 year-old child 10 days (A, B, E, F, I, J, L, M) and 56 days (C, D, G, H, K, N, O) after the onset of FM. He was hospitalized after four days of a viral syndrome followed by acute hemodynamic collapse with 38% of LVEF. Increased myocardial thickness and normal LVEDD was found in the end-diastolic (A, E) and end-systolic cine images (B, F) at baseline CMR. At follow-up CMR, myocardial thickness returned to normal (C, D, G, H). Diffuse myocardial edema was shown in the T2-weighted images with the T2 ratio of 1.98 at baseline CMR (I, J) and disappeared at follow-up (K, T2 ratio:1.64). Regional mid-wall LGE within the anteroseptal and inferoseptal walls were found at baseline CMR (L, M) and disappeared at follow-up (N, O). CMR, cardiac magnetic resonance; FM, fulminant myocarditis; LVEF, left ventricle ejection fraction; LVEDD, left ventricular end-diastolic diameter; LGE, late gadolinium enhancement.

of clinical manifestations, immunological features and CMR findings could be found in most FM children. The presence of III° AVB and smaller LGE areaat baseline CMR might indicate better short-term outcomes.

In our study, positive viral antibodies were observed in children, while the viral serology was not applied in the diagnosis of FM. Positive viral serology does not imply myocardial infection but rather indicates the interaction of the peripheral immune system with an infectious agent (*Caforio et al., 2013*). The prevalence of circulatory viral antibodies (IgG) was also found in health children. Besides, no correlation between virus serology and EMB findings was found (*Mahfoud et al., 2011*).

In our study, myocardial thickness increased in 75% FM children at baseline CMR and returned to normal at follow-up. Similar findings were also found in several previous studies. *Felker et al. (2000)* found increased septal thickness in 11 FM patients at presentation by echocardiography and normal septal thickness six months later, compared with normal septal thickness in acute myocarditis (AM). Felker attributed the increased myocardial thickness to inflammatory response seen on EMB. Severe LV wall thickening was also reported in a 22-year-old FM woman with cardiogenic shock (*Shillcutt et al., 2015*). In our study, we also found the correlation between increased myocardial thickness and

myocardial edema. At baseline CMR, all FM children with increased myocardial thickness performed myocardial edema. At follow-up CMR, the myocardial thickness returned to normal, with only one myocardial edema left. Increased myocardial thickness might be the response of myocardial inflammation and could be a characteristic radiological finding to distinguish FM from AM.

In our study, increased or normal LVEF ($71.8 \pm 6.9\%$) was found in all children at baseline CMR of FM, compared with impaired LVEF ($36.5 \pm 8.2\%$) within 72 h of onset. The LV cardiac function returned to normal ($64.9 \pm 7.1\%$) at follow-up. Clinical interventions and treatment undoubtedly played a primary role in the recovery of LV systolic function (*Ginsberg & Parrillo, 2013*; *McCarthy et al., 2000*). While we thought that increased LVEF might also be associated with increased myocardial thickness. At baseline CMR, increased myocardial thickness could be found in 75% children, leading to significantly decreased ESV ($12.9 \pm 5.0$ VS $18.2 \pm 7.4$ ml, $P = 0.016$) and little change of EDV ($44.6 \pm 12.6$ VS $50.6 \pm 16.5$ ml, $P = 0.161$), compared with those at follow-up. Then, LVEF increased according to the formula:

$$LVEF = (EDV - ESV)/EDV.$$

The ESV in children was more vulnerable to be affected by myocardial thickness because of their small LV volume.

One of the advantages of CMR was the noninvasive evaluation of myocardial tissue characterization. In our study, myocardial hyperemia was shown in 100% children, followed by myocardial edema and necrosis/fibrosis (87.5%), which indicated more severe inflammatory response. The CMR diagnostic sensitivity for FM at baseline CMR was considerably high (100%). Endomyocardial biopsy (EMB) had confirmed that "active myocarditis" was more common in FM than that in AM, which indicated higher degree of myocardial inflammation in FM (*Felker et al., 2000*). In AM with "borderline myocarditis," myocardial tissue characterization might not be observed because of less severe inflammation (*Friedrich et al., 2009*; *De Cobelli et al., 2006*). While in FM with more common "active myocarditis", CMR tissue characterization could be a useful tool for FM diagnosis.

The exact pathophysiology of LGE in myocarditis is still under investigation. Myocardial inflammation and/or necrosis in the acute phase seemed to play a major role (*Mavrogeni et al., 2012a*). High specificity of LGE for the detection of myocardial injury in myocarditis had been demonstrated in several studies (*Cooper et al., 2007*; *Abdel-Aty et al., 2005*; *Mahrholdt et al., 2004*). In our study, patchy mid-wall or subepicardial LGE in the LV walls were found in seven FM children (87.5%), which were similar to those in AM children (*Banka et al., 2015*; *Sachdeva et al., 2015*). Similar mid-wall LGE were also found in a 33-year-old female FM patient (*Ryu et al., 2013*). While *Takeuchi et al. (2010)* and *Mavrogeni et al. (2012b)* found normal LGE in their reports. One of the possible reasons for this might be the longer intervals from onset to CMR examinations (10 days, 7–20) in our study. Besides, the differences of FM between adult and children need further study.

The outcomes of FM in the pediatric population were controversial *Amabile et al. (2006)* found that 90.9% FM children performed favorable outcomes, with no symptoms and

normalized LVEF. While in a nationwide survey of Japanese children and adolescents reported by *Saji et al. (2012)* showed that the survival rate for children with fulminant MC was disappointing (51.6%). In our study, full recovery of clinical manifestations and cardiac function could be found in all FM children at follow-up. Myocardial inflammation disappeared completely in 62.5% FM children. *Ryu et al. (2013)* reported a 33-year-old female FM patient with increased T2WI signal and LGE and these findings disappeared 3 months later. While for three children with potential myocardial inflammation, myocardial edema had lightened (T2 ratio, 1.96 ± 0.22 VS 1.90 ± 0.08) and LGE areas were smaller (138.0 ± 25.2 mm$^2$ VS 95.0 ± 26.0 mm$^2$). We thought the still presences of myocardial inflammation were due to short intervals of follow-up CMR (median, 54 days).

In our study, we found the presence of III° AVB at baseline CMR of FM might indicate better short-term outcomes. The favorable outcomes associated with complete AVB in FM and AM were also reported. *Lee et al. (2014)* reported the short-term outcomes of FM children in a single center and found that all seven FM children with complete AVB on the initial EKG survived. The AM patients with complete AVB perform excellent survival rates (89–100%) and long-term outcomes (100%). 11–28% AM patients had persistent complete AVB at hospital discharge and became normal at follow-up (*Chien et al., 2008*; *Wang et al., 2002*; *Batra, Epstein & Silka, 2003*). The reason why FM children with complete AVB had good outcomes was unclear. We presumed that because of the more specific manifestations of myocarditis with CAVB, such as hypotension and Stokes-Adams seizures (*Chien et al., 2008*), patients could receive earlier treatment before the development of obvious congestive heart failure (*Wang et al., 2002*).

Smaller LGE area at baseline CMR of FM also indicated better short-term outcomes. The value of CMR in the assessment of FM outcomes had not been reported. CMR has been applied to predict the outcomes of AM. *Sachdeva et al. (2015)* reported that LGE could be found in half of AM children and it could be identified as predictors of poor outcomes. *Barone-Rochette et al. (2014)* also applied a simplified visual quantitative score of LGE to identify the outcomes of AM patients, with LGE identified as an independent predictor of all cause and cardiac mortality in these AM patients. In patients with AM, significantly continuous decrease of LGE volume percentage had been observed over several follow-up CMR examinations, which demonstrated a rapid and continuous decrease of myocardial inflammation (*Luetkens et al., 2016*). The myocardial inflammation was easier to subside in a small area of LGE, which indicated better outcomes.

Nowadays, T1 and T2 myocardial mapping techniques have been applied in the assessment of myocarditis as novel quantitative tissue markers (*Moon et al., 2013*; *Hamlin et al., 2014*; *Radunski et al., 2014*; *Lurz et al., 2016*). The extracellular volume (ECV), as calculated by the pre- and post-contrast T1 values and HCT, could directly and non-invasively measure the proportion of extracellular space within the myocardium. Compared with Lake Louise Criteria, significantly improved diagnostic accuracy had been reported in patients with myocarditis using mapping techniques (*Radunski et al., 2014*; *Lurz et al., 2016*). *Lurz et al. (2016)* reported that mapping techniques could provide a useful tool for the diagnosis of acute myocarditis and were superior to the LLC. *Radunski et al. (2014)* found that ECV combined with LGE imaging could significantly improve the diagnostic accuracy of CMR

compared with standard Lake Louise Criteria in patients with severe myocarditis. In our study, Lake Louise Criteria was applied in the CMR diagnosis of FM. The diagnostic accuracy of mapping techniques would be discussed in the future.

This study has several potential limitations. First of all, the number of patients included in the study was small. While all children performed CMR examinations twice at different course of FM, the CMR findings were convictive by comparing at baseline and follow-up CMR. Second we have utilized clinical rather than histologic criteria in making the diagnosis of FM. Since the clinical manifestations and CMR findings of our patients were consistent with FM and full recovery of cardiac function was found with supportive care, the diagnosis of FM was definite. Third, the EGE examinations were performed in 62.5% FM children at baseline CMR and only one child at follow-up. As a result, we could not evaluate the recovery of myocardial hyperemia. Fourth, the intervals from onset to CMR examinations were varied based on patient's conditions, which might had an influence on the CMR findings. Fifth, the CMR protocol in 8 children was different because multiple signal averaging were applied in the cine imaging and LGE imaging of two small children under free-breathing condition. In our study we focused on the CMR differences at baseline and follow-up. The CMR protocol at baseline and follow-up in each child was same and paired $T$-test was used. We didn't compare the CMR findings between free-breathing and breath-hold children. Therefore, the protocol difference had no influence on our results. Sixth, because of the small sample size, the clinical data and initial CMR findings between FM children with different short-term outcomes were directly compared. The predictors of FM outcomes were not demonstrated by the statistical analysis but by the differences between the two groups that are described.

## CONCLUSIONS

The CMR findings of FM in children were characteristic and useful for early diagnosis. Full recovery of clinical manifestations, immunological features and CMR findings could be found in most FM children. The presence of III° AVB and smaller LGE area at baseline CMR might indicate better short-term outcomes.

## ACKNOWLEDGEMENTS

We would like to acknowledge the participation of the study patients and their families. We also wish to acknowledge the support of the Department of Pediatrics.

### Funding

This study was supported by the Shandong Provincial Natural Science Foundation of China (grant nos. ZR2012HM042, BS2015YY003) and Shandong Provincial Medical and Healthy Technology Development Program of China (grant nos. 2015WS0176). The funders had no role in study design, data collection and analysis, decision to publish, or preparation of the manuscript.

## Grant Disclosures

The following grant information was disclosed by the authors:
Shandong Provincial Natural Science Foundation of China: ZR2012HM042, BS2015YY003.
Shandong Provincial Medical and Healthy Technology Development Program of China: 2015WS0176.

## Competing Interests

The authors declare there are no competing interests.

## Author Contributions

- Haipeng Wang performed the experiments, analyzed the data, wrote the paper, prepared figures and/or tables.
- Bin Zhao conceived and designed the experiments, reviewed drafts of the paper.
- Haipeng Jia performed the experiments, wrote the paper.
- Fei Gao contributed reagents/materials/analysis tools, reviewed drafts of the paper.
- Junyu Zhao analyzed the data, contributed reagents/materials/analysis tools.
- Cuiyan Wang conceived and designed the experiments, performed the experiments, analyzed the data, reviewed drafts of the paper.

## Human Ethics

The following information was supplied relating to ethical approvals (i.e., approving body and any reference numbers):

The study protocol was approved by the institutional ethics committee of Shandong Medical Imaging Research Institute (NO. 2016-001).

## Data Availability

The raw data has been supplied as Supplementary File and is available at FigShare: Wang, Haipeng (2016): the CMR findings of FM. Figshare.
https://dx.doi.org/10.6084/m9.figshare.3515498.v1.

## Supplemental Information

Supplemental information for this article can be found online at http://dx.doi.org/10.7717/peerj.2750#supplemental-information.

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
