# Peer review of "A retrospective study: cardiac MRI of fulminant myocarditis in children—can we evaluate the short-term outcomes?"

_PeerJ, doi:10.7717/peerj.2750_

## Round 0.1 · original submission · Major Revisions

· Academic Editor

Major Revisions

Dear Authors, the reviews are enclosed. Please reply to all the points raised and re-elaborate the manuscript accordingly.

·

Basic reporting

No Comments.

Experimental design

No Comments.

Validity of the findings

No Comments.

Comments for the author

Published pediatric data evaluating clinical parameters and cardiac MRI findings in children with fulminant myocarditis are sparse. Therefore, the current article integrates the existing knowledge of this very specific patients category with fulminant myocarditis.
Overall, the article is well written and easily to read. Tables and Figures are well presented.

Results
- Page 12, Lines 168-169. Please, add the reference limit of the Troponin.
- Did the patients underwent specific viral studies? And if not, why not? Please, specify that in the text.

Discussion
- The authors should analyze and explain the text the observations that the higher incidence of III° AVB and smaller LGE area at acute/subacute stage indicate better short-term outcomes.
- The authors should consider a paragraph regarding the T1/T2 mapping MRI sequences that may be an evaluable new tool in assessing global myocardial alterations in acute myocarditis by providing quantitative T1 or T2 values. Furthermore, the ECV fraction, as calculated by the pre-and post-contrast T1 values and the patient's hematocrit, can provide information regarding the presence or absence of extracellular expansion of the myocardium.

Reviewer 2 ·

Basic reporting

- There is the need for a native English-speaker to review and edit the paper.
- The Discussion section is not so clear and not so easy to understand.
- Figure 2: J: please report the RATIO; K-L: these images don’t seem to be useful, please remove them or explain their added value; M-N: the LGE is not so clear. Please adjust the window level to improve the LGE visualisation.

Experimental design

- Please include the fulminant miocarditis definition in the text (eg: Gupta S, Markham DW, Drazner MH, Mammen PP. Fulminant myocarditis. Nat Clin Pract Cardiovasc Med. 2008 Nov;5(11):693-706. doi: 10.1038/ncpcardio1331. Epub 2008 Sep 16. Review. PubMed PMID: 18797433).
- MR scans were performed after the “ventricular function recovery” as stated by the Authors (LVEF 36.5±8.2% at presentation; LVEF: 71.8±6.9% at AS CMR findings). So, I’m not sure if it is correct to talk about Acute MR evaluation.
- Materials & Methods line 94: “Children who could control breathing (>7 years) would acquire data with respiratory gating. Small children (<7 years) would be sedated with 10% chloral hydrate and examined under free-breathing condition”. The median age of eight FM children was 8.5 years old (range, 3 to 14). Please, a better explanation of the protocol and differences in results are required.

Validity of the findings

- Results line 213-216: please try to explain the presence of III AVB in patients with small LGE at acute/subacute stage of FM.
- Discussion line 224-229: please explain the relation with your work or remove it.
- Discussion lline 270-271: “LGE specifically reflects irreversible myocardial injury (necrosis and fibrosis) caused by more severe inflammation”: this sentence is not clear respectively to your data.
- please highlights in the Limits section that prognostic data are not demonstrated by your analysis but the differences between the two groups are only described.

---

## Round 0.2 · accepted · Accept

· Academic Editor

Accept

The revision has been satisfactory and there are no further comments.

Reviewer 2 ·

Basic reporting

nothing to disclose

Experimental design

nothing to disclose

Validity of the findings

nothing to disclose

Comments for the author

The changes have substantially improved the manuscript.